# Crack Detection in Frozen Soils Using Infrared Thermographic Camera

**DOI:** 10.3390/s22030885

**Published:** 2022-01-24

**Authors:** Yang Zhao, Yufan Han, Cheng Chen, Hyungjoon Seo

**Affiliations:** 1Department of Civil Engineering, Xi’an Jiaotong Liverpool University, Suzhou 215000, China; yang.zhao@xjtlu.edu.cn (Y.Z.); cheng.chen19@student.xjtlu.edu.cn (C.C.); 2Department of Computer Science, Xi’an Jiaotong Liverpool University, Suzhou 215000, China; Yufan.Han20@student.xjtlu.edu.cn; 3Department of Civil Engineering and Industrial Design, University of Liverpool, Liverpool L69 7ZX, UK

**Keywords:** frozen soil, infrared camera, crack

## Abstract

Frozen soils are encountered on construction sites in the polar regions or regions where artificial frozen ground (AFG) methods are used. Thus, efficient ways to monitor the behavior and potential failure of frozen soils are currently in demand. The advancement of thermographic technology presents an alternative solution as deformation occurring in frozen soils generate heat via inter-particle friction, and thus a subsequent increase in temperature. In this research, uniaxial compression tests were conducted on cylindrical frozen soil specimens of three types, namely clay, sand, and gravel. During the tests, surface temperature profiles of the specimens were recorded through an infrared video camera. The thermographic videos were analyzed, and subsequent results showed that temperature increases caused by frictional heat could be observed in all three frozen soil specimens. Therefore, increases in temperature can be deemed as an indicator for the potential failure of frozen soils and this method is applicable for monitoring purposes.

## 1. Introduction

Historically, humans lived and built structures on frozen ground among Arctic areas where annual average temperatures were below the freezing point of water. Artificial frozen ground (AFG) methods have recently gained popularity as a technique to stabilize soil during excavation. Due to the bonding effects of ice, frozen soil is stronger and more rigid than regular soil and may demonstrate mechanical behavior similar to concrete. The crack initiation and propagation of concrete has been studied by numerous researchers, but few studies have focused on cracks in frozen soil.

As cracking occurs, there is a relative movement of soil particles around the crack, and hence frictional heat. The frictional heat may cause temperature increases, which can be observed using a thermographic device such as an infrared camera. Researchers successfully used an infrared camera to detect various phenomena with temperature changes, such as oil products spreading on water surfaces [1], wild fires [2], and wind flow [3]. Specifically, the thermographic technique was applied to structural defects. The behavior of rock and soil was determined through monitoring with infrared thermal cameras [4,5]. Liu, et al. [6] applied infrared monitoring in an experimental study of a tunnel. Seo, et al. [7], Seo [8] detected crack formation in pillars using an infrared camera. Moreover, an infrared thermographic camera was applied in ice detection on wind turbine blades [9] and aircraft air foils [10].

Though in some cases, the defects fail to generate temperature changes when the structure is at rest. However, when heated by an external source, defects will display different temperature compared with the rest of the structure. Broberg [11] used an infrared camera to detect welding defects based on the temperature difference between defects and surrounding surfaces while the weld was heated by a flash lamp as an external IR source. Štarman and Matz [12] observed the propagation of artificially generated thermal pulses in steel bars using an infrared camera to locate the presence of cracks. Afshani, et al. [13] conducted a study to detect defects in the lining of a tunnel with an infrared thermal camera. Recently, deep learning and machine learning analyses were applied to find a crack in infrastructures [14,15]. Moreover, the three-dimensional monitoring system was applied for monitoring the displacement and tilt of infrastructure using laser scanning [16]. In this paper, an experimental study was conducted by simulating frozen soils. The cracks of different frozen soils were identified through infrared topography.

## 2. Methods

### 2.1. Specimen Preparation

Three cases of soil category were considered in this study, namely clay, sand, and gravel. For the clay specimen, an undisturbed clay sample extracted from a depth of 3 m was cut into a cylinder of 100 mm (diameter) by 150 mm (height). Next, the sample was submerged in a water tank for 24 h until reaching saturation. Afterward, it was placed in a freezer for at least 24 h. For the sand and gravel specimens, soil particles and water were poured into molds iteratively to ensure the specimen was fully saturated and composition was relatively uniform across the height. Then, they were frozen in a freezer before being removed from the mold. Subsequently, the specimen was again stored in the freezer for 24 h before the test. Figure 1 shows the process of forming the frozen sand specimen. Due to mechanical disturbances during the removal of the specimen, the height of the sand and gravel specimen was not strictly controlled. The properties of the specimens is shown in Table 1 and photos of specimens were taken before uniaxial compression tests as shown in Figure 2.

### 2.2. Thermal Graphic Imaging

Temperature changes can be detected remotely and non-intrusively. Additionally, the thermograph provides 2D geometric information. Therefore, an infrared camera was used in this research to detect crack formation in frozen soils. The infrared camera used was the FLIR E60. The exported results file consisted of sequences of thermographic photos taken at a frequency of 30 Hz and resolution of 320 * 240 pixels. Each pixel in a frame had its temperature measured. The exported thermographic files were analyzed using the software FLIR Research IR. The thermal sensitivity was 0.05 °C and the accuracy was ±2 °C.

All bodies above 0 K (Kelvins) emit electromagnetic radiation. The characteristic of the radiation depends on the temperature of the body, thus the temperature of the body can be determined by the measured radiance. Infrared is a section of electromagnetic radiation with wave length ranging from 780 nm to 1 mm [17]. At room temperature, the majority of radiation energy lies within the range of infrared, which makes an infrared sensor the ideal selection for the purpose of temperature measurement. When an object is photographed using an infrared camera, the reflected infrared radiation emitted by the object is received by the infrared camera’s sensor and simultaneously converted to an electronic signal. Then, the electronic signal is processed by the controlling software to generate a thermal graphic image or, in short, a thermograph. 

The governing equation for the relationship between thermal radiation intensity and temperature is shown by Stefan–Boltzmann’s law (1)
(1)$$W=\epsilon {\sigma T}^{4}$$
where W is radiance intensity ($W/{\;m}^{2})$ measured by the infrared sensor, Stefan–Boltzmann constant$\;\sigma =5.67\times 10^{-8}\;\frac{W}{m^{2}\cdot K^{4}}\;$, T is the temperature of the object (K), ϵ is emissivity. In this research, software FLIR Research and Design [18] was used to process the infrared measurement and the back-calculation from the measured radiance intensity to temperature based on Equation (1).

Ideally, to obtain the precise relationship of values W and T, it is necessary that we quantify the values of ϵ precisely at any given time and location, which requires robust and sophisticated calibration and strict control over atmospheric conditions. Such requirements are practically difficult and unnecessary if the sole purpose concerns the occurrence of abnormal changes in temperature rather than measuring the exact value of the temperature. In this research, ϵ=0.95 is assumed.

### 2.3. Test Procedures

In this research, uniaxial compression tests were conducted on frozen soil specimens while the specimens were subjected to an infrared camera. The arrangement of the experimental apparatus is shown in Figure 3. 

The infrared camera was placed approximately 1 m in front of the compression machine. The load was recorded at a frequency of 10 Hz. The T-load curve was visible simultaneously on laptop 2 controlling the compression machine. The uniaxial compression test was terminated when the specimen failed or the lift of the piston reached its capacity.

## 3. Result and Analysis

Tests were conducted on 6 samples for each soil type where multiple samples for each soil type can be seen in Figure 1. However, we were unable to observe the cracking behaviors on all of the tests using an infrared camera due to the following three conditions: (a) cracks occurred on the backsides of the samples, (b) samples disintegrated without cracking due to melting (see Figure 4a,b), and (c) samples yielded without cracking (see Figure 4c). Cracking occurred on either one or both sides of the specimen by chance, which cannot be practically controlled. Yielding and disintegrating of specimens occurred as the specimens were partially melted. The authors indeed attempted to repeat the tests under a lower room temperature (around 15 °C) in the winter, however, satisfactory improvements on avoiding the melting of specimens were not achieved. To avoid the melting of specimens, the compression tests on specimens should be conducted under temperatures below the freezing point of water [19,20,21]. In the previous research [19,20,21], the low temperature of specimens during testing was maintained by immersing the specimens in liquid cool materials. Unfortunately, such cooling systems were not available to the authors and restricted the direct observation of specimens using the infrared thermographic camera during compression tests. Therefore, specimens were not guaranteed to be completely frozen during our tests. One sample of each soil type was analyzed as they demonstrated evident cracking behavior.

Due to a lack of pre-existing knowledge of conditions under which a crack would appear in thermograph of frozen soils, the thermographs were carefully examined frame by frame. The qualification of a crack can be qualitatively described as follows:(a)Significant temporal temperature variation, compared to the immediate surrounding area (baseline), occurring at certain spots of a relatively small area.(b)The temperature variation initiates within the specimen surface and is not transmitted from the interface between the specimen and the ambient environment.(c)The temperature variation is sustained for more than 1 frame, which disqualifies false positives caused by random flocculation in the measured value of temperature.(d)The temperature variation is not caused by mass transportation i.e., movement of disintegrated soil particles or water flow thawed from ice.

According to the above description, two types of curves are plotted to demonstrate the temperature change at a crack point. The T-t curve shows how the temperature at the crack point varies temporally. The T-d curve shows how the temperature at the crack point varies spatially. Moreover, the load-time curve is used to qualitatively determine the strain-stress status of the specimen. It was assumed that the frozen soil specimens displayed elastic-plastic behavior. The period before the load peaks is defined as the elastic stage and the period after as the plastic stage. To convert the unit of distance measured in the thermograph from pixel to mm, the thickness of the lower platen of the compression machine was used to calculate the conversion ratio from pixel to mm. The calculated conversion ratios for frozen clay, sand and gravel were 0.60, 0.62, and 0.65 mm/pixel, respectively.

### 3.1. Crack in Frozen Clay

As shown in the load-time curve in Figure 5, the load peaked at 96.30 s after the test commenced. After 96.30 s, the temperature of certain points on the specimen’s surface appears to increase at a slightly higher rate than the rest. One example is indicated in Figure 6 as the crack point. A baseline point about 4 mm downwards of the crack point was selected to mitigate the effect of measurement errors and the effect of heating from the ambient atmosphere on the variation of temperature. The ΔT (temperature change)-t (time) curve for the crack point and baseline points are plotted in Figure 7. The temperature profile was set relative to temperature changes, with 0 chosen as the initial temperature at the reference point. The temperature changes were filtered by implementation of a Savitzky–Golay filter.

Disregarding the random flocculation, from 105 s to 123 s, the temperature increases from −3.02 °C to −2.41 °C. 

The temperature increase at the crack point was further verified by the temperature profile along a line approximately perpendicular to the crack as shown in Figure 8. The temperature profile at the start of the test (0 s), immediately before the crack occurs (105 s), and after the crack forms (125 s), are plotted. Before the crack occurs, there is no significant difference in temperature between the potential crack point and the rest. After the crack is formed, the temperature at the crack point is 0.57 °C higher than those at the points not influenced by the crack. 

### 3.2. Bulge Effect in Frozen Sand 

Although there was no individual crack observed in the sand case, as the frozen sand specimen was compressed, the expansion in the radial direction, which we termed bulge, became significantly visible after yield and the observation can also be correlated to temperature variations. The temperature of three representative points located at the upper part (U.), lower left part (L.L), lower right part (L.R), and the average temperature of a square area (S.) were selected to demonstrate the effect of the bulge on temperature. The locations are indicated in Figure 9. In Figure 9, temperatures are mostly above 0 °C as the surface of the specimen was covered by water instead of ice. The ambient air temperature was around 30 °C. The ice on the top of the specimen melted immediately after contact between the specimen and upper platen of the compression machine. Due to gravitational effects, the water subsequently ran off downwards onto the specimen surface. Although the specimen was completely frozen initially, certain parts melted during the compression test. The same observation was made for the clay specimen and gravel specimen.

The temperature and load curves are plotted together in Figure 10. As the bulge of the specimen becomes evident after yield (64.9 s), the temperature of each single point of the specimen surface demonstrated 4 simultaneous impulsive increases initiating at 66.80 s, 88.30 s, 99.53 s, and 110.30 s, respectively. Impulses were not observed at points outside of the specimen surface, which signifies these impulses were not measurement errors. The magnitude of ΔT for the impulses ranged from 0.37 to 0.95 °C.

### 3.3. Cracks in Frozen Gravel

Two types of cracks were observed during the compression test of the gravel specimen. The cracks, which occurred in-between gravel particles, caused increases of temperature at the crack point. Such cracks were denoted as I1, I2 I3, and I4 for ‘increase’. Temperature decreases at the crack point were observed for cracks which occurred within the ice block. Such cracks were denoted as D1, D2, D3, D4, D5, and D6 following the capitalized initial of the word ‘decrease’. The ideal temperature variation for cracks of types I (increase) and D (decrease) are illustrated in Figure 11. When cracks occur between gravel particles, the inter-particle friction generates heat. However, those frictional forces were relatively negligible between ice surfaces due to their smoothness. Although there was heat generated around ice particles, the heat would probably be consumed by the melting of ice as opposed to an increase in temperature. The temperature decrease is due to a temperature gradient travelling from the surface to the inner core of the specimen. As the crack widens, the inner surface of the specimen, the temperature of which is lower compared to the outside, is exposed to the camera. The location of these two types of cracks is indicated in Figure 12.

Temporal variations for cracks I1, D2 and D5 are shown in Figure 13a, Figure 13b and Figure 13c, respectively. Crack I1 occurs at the time of yield. Before the crack initiates, the temperature of the crack point is relatively constant and approximately equal to the baseline temperature. The ΔT-t curve of the crack point deviates from that of the baseline since t = 126.33 s. From t = 126.33 to t = 128.80 s, the temperature at the crack point increases by 0.6 °C while that at the baseline remains relatively stable. After the crack is formed, the temperature at the crack remains relatively steady from 128.90 to 135.33 s but subsequently decreases due to the decrease in load, as shown in Figure 14.

Crack D2 forms at the transitional point between stress softening and plastic deformation, which takes less time than the formation of crack I1. This is compatible with the explanation that there is minimal friction and ice is extremely brittle The magnitude of the temperature decrease is less important than the temperature increase for the crack between gravel particles as the decrease only demonstrates the depth of the crack that develops into the specimen. Crack I5 occurs along the global failure surface during plastic deformation. The crack occurred at 265 s with a temperature increase of 0.51 °C. Following the relative movement between two parts of the specimen on each side of the failure, the surface starts at a 280 s to display a temperature increase that takes 3 steps, which corresponds to the steps of relative movement along the failure surface observed in the video. The load, and hence the stress within the specimen, is less than those observed previously during the elastic stage when I1 occurs. However, the relative movement along the global failure surface is more intense. Thus, both the rate of increase and the total amount of increase in temperature is larger. From the 280 s to 315 s, the temperature increase due to friction along the global failure surface is 2.66 °C. The spatial temperature distribution across I1, D2, and I5 are also plotted to demonstrate the increase during the formation of the crack in Figure 15. $\Delta T_{I1}$ is defined as
(2)$$\Delta T_{I1}=T_{I1}-T_{average}$$
where $T_{I1}$ is the temperature at crack I1 at formation and $T_{average}$ is the average at the immediate surrounding area of crack I1.

In the same manner, the time, $t_{i}$, represents when the crack is fully developed and $\Delta T_{i}$ is recorded for cracks I2, I3, I5, D1, D3, D4, and D6. After a crack has formed, the temperature may still vary, subject to the change in stress-strain conditions. Thus, the ΔT for crack I1 at the stress-softening stage (denoted by the initial ‘s’ as subscript in $I1_{s}$) and plastic stage (denoted by the initial ‘p’ as subscript in $I1_{p}$) and ΔT for Crack I2 at the plastic stage were also calculated.

The series of $(t_{i}$,$\;\Delta T_{i}$) are plotted together with the load-time curve in Figure 16. During the elastic stage, only two cracks, I1 and D1, occur slightly before yield (128.9 s). As the load decreases during the stress-softening period, the ΔT at crack I1 also decreases. A series of cracks consisting of 4 types D and 1 type I occur immediately before the transition to plastic deformation at 162.0 s. Afterward, the load remains relatively constant as the compression reaches its plastic stage and no crack occurs until t = 265 s when the cracks later form the initiating global failure line. From t = 265 s to t 300 s, the global failure surface gradually grows by connecting the individual cracks and the temperature along with it generally increases. The total increase in temperature at these series of cracks are all above 2.1 °C.

Figure 17 shows the temperature profile and normalized temperature profile around I1 at yield, stress softening and failure. ΔT is a result of frictional heat but its magnitude also depends on the thermal properties. It is assumed that the thermal properties and conditions of crack I1 remain constant during the test.

For the crack I1, values μ and C can be assumed constant during the test. Although the stress at the crack was not measured, it can be inferred that the stress is proportional to the load. Before the plastic stage, the deformation of the entire specimen was relatively uniform and there was little inter-particle movement. There, the ΔT is mostly determined by the load. The load is the highest at yield and subsequently decreases during the stress-softening stage while the rate of displacement shows no significant change throughout the period. Thus, $\Delta T_{y}$ is higher than $\Delta T_{s}$. After the plastic stage was reached, although the stress is lower than before, the rate of displacement is significantly higher. The $\Delta T_{p}$ continues increasing and, when the specimen eventually fails,$\;\Delta T_{p}$ is much higher than $\Delta T_{y}$ and $\Delta T_{s}$.

With the normalized profile, the propagation of frictional heat can be examined. The dimension affected by the frictional heat expands during the test but is ultimately limited by the dimensions of the gravel particle directly subjected to frictional forces. It seems the temperature of ice within the void of particles remains unaffected by the frictional heat. There are two possible explanations:The ice has a much higher heat capacity than the gravel.The heat transferred to the ice will be consumed by the melting of ice rather than causing an increase in temperature.

### 3.4. Comparison of Behavior of Plastic Stage

In Figure 18 the spatial temperature profile of cracks from all 3 cases are plotted together. The maximum value of ΔT is observed in Figure 17 also plotted together with the load curve in Figure 19a. The strain is estimated from the deformation of specimen in the videos. However, the magnitude of ΔT cannot provide an undistorted view of the magnitude of friction as the ΔT also depends on thermal parameters such as the specific heat capacity besides the frictional heat. Thus, the frictional heat is back-calculated from ΔT.

A body of unit amount of mass is considered for each specimen and the body is assumed as an isolated system. Regarding the clay and gravel specimens, the body is located in the pixel referred to as the crack point. For the sand specimen, the body is located in an arbitrary position within rectangle S. Thus, the amount of frictional heat that occurred in those bodies of unit weight can be calculated as:(3)$$\Delta H=C_{m}\times \Delta T$$
where $C_{m}$ is the specific heat capacity of unit mass, ΔT is the temporal temperature change resulting from crack or bulge. The value of $C_{m}$ is obtained from the literature as shown in Table 2.

As the spatial resolution was about 0.4mm/pixel in this study, for the calculation of the gravel specimen, pixels of the crack point observed were within the gravel particle, thus the $c_{m}$ for dry gravel was directly used. For the clay and sand specimens, the $c_{m}$ for the soil–water mixture should be used. The $c_{m}$ for sand is very much dependent on the void ratio of the sand specimen. The $c_{m}$ for the sand–ice mixture is assumed to be the weighted average of specific heat capacity for ice and dry sand and calculated as
(4)$$C_{m}=\frac{C_{s}\times m_{s}+C_{i}\times m_{i}}{m_{s}+m_{i}}$$
where $C_{s}$ is the specific heat capacity per unit mass of sand, $C_{w}$ is the specific heat capacity per unit of ice, $m_{s}$ and $m_{w}$ are the mass of sand and ice(water) constituting the specimen, $C_{s}=0.770\;J/\left(\mathrm{gK}\right)$ is cited in Table 2, $m_{s}=2257$ g and $m_{w}=370\;g$ are measured when the specimens are prepared. $C_{i}$ is estimated according to an empirical equation proposed by Dickinson and Osborne [24] as follows:(5)$$C_{i}=2.114+0.007789T$$

The value of T (temperature) should be assigned to the unit of K. To be consistent with the heat capacity for gravel and clay measured at 0 °C, T = 273.15 K is used in Equation (4), $C_{i}=$2.116 J/(gK). With all the values on the right-hand side in Equation (3) obtained, $C_{m}$ can be calculated. The values of $C_{m}$ adopted for all three cases are listed in Table 3 and these values were used in Equation (4) to calculate the ΔH for each case.

Figure 19b shows the ΔH in unit mass resulting from inter-particle friction and load–strain curve for all 3 cases altogether. As shown in Table 3, the equivalent unit weight specific heat capacity $C_{m}$ of frozen gravel is the lowest among the three different specimens. It is assumed the frictional heat generated is proportional to the work done by the compression. The work done by compression is approximately proportional to the product of load and axial strain in Figure 19. Before the maximum load points of the specimens, the gravel specimen experienced the most frictional heat ΔH. Therefore, according to Equation (4), $\Delta T=\Delta H/C_{m}$, the temperature change of gravel is higher than that of sand and clay.

For the clay specimen, during the plastic deform, there was significant radial deformation at the top and bottom of the specimen, which occurred immediately contact of the platen of the compression machine. Moreover, the rest of specimen deformed similar to an extremely viscous fluid. The friction results from the differential displacement rates between neighboring clay particles. For the sand specimen, as the specimen bulges, the sand particle layers must rearrange. Particles are squeezed into neighboring layers in an axial direction and hence friction occurs. As the friction is distributed among the entire specimen, thus ΔH in unit mass is the least among the 3 cases during plastic deformation. For the gravel specimen, the ΔH is more than 2 times that of the other two cases, because the friction concentrates on the global failure surface where two parts of the specimen slide against each other. The same set of curves except for that of the sand case are also replotted in Figure 20 with normalized ΔT. This shows that the affected distance of the crack in the clay case is less than that of the gravel case. This might be related to the particle size of the soil. As can be seen in the case of gravel, the surface of the temperature rise propagates through the ice in the voids of the soil particle skeleton. Moreover, the estimated specific heat of gravel was less than half of that of frozen clay. This reveals that it took less amount of heat for the frozen gravel to increase a unit of temperature than for the frozen clay. Thus, the temperature increase due to frictional heat occurred in a smaller spatial dimension in frozen clay than in frozen gravel.

## 4. Discussion

The identification of the change in thermographic profile was labor intensive and difficult to manage in real time. Thus, the application of this method for monitoring purpose on-site requires significant improvements in efficiency in the future. For example, deep-learning image pattern recognition methods could be applied to automate the process of identification of cracking patterns in thermography.

In this research, tests were conducted at summer room temperature conditions (between 25 °C and 32 °C) and thermodynamic processes occurred, such as heat transmission between specimens and platens of compression machine occurred. It was assumed that temperature changes were not affected by thermodynamic processes. Emissivity of specimens were assumed constant and uniform among all samples. Both assumptions may not agree with real-life conditions upon application of this method in the field.

## 5. Conclusions

In this research, the thermographic profile of frozen soil specimens under uniaxial compression test was studied. For all 3 cases, abnormal temperature variations on specimen surfaces due to inter-particle friction occurred were observed.

In frozen clay specimens, the temperature increases at cracks were identified only after plastic deformation occurred.For frozen sand, simultaneous temperature increases were observed along the entire specimen as it bulged at the plastic stage.In frozen gravel, temperature changes were observed for before cracks appeared in addition to a yield. For cracks in ice particles, there were temperature decreases due to changes in geometry. For cracks in gravel particles, there were temperature increases due to inter-particle friction.Compared in terms of ΔH in unit mass it was shown that the friction in gravel was the strongest and that in sand was the least.The propagation of temperature increases from cracks were also examined in frozen clay and gravel cases. Subsequently, it was shown that the temperature rise propagates further in gravel than in sand.

## Figures and Tables

**Figure 1 sensors-22-00885-f001:**
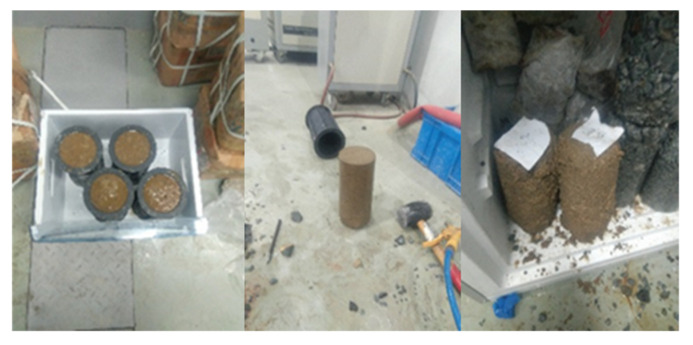
Preparation of sand specimen.

**Figure 2 sensors-22-00885-f002:**
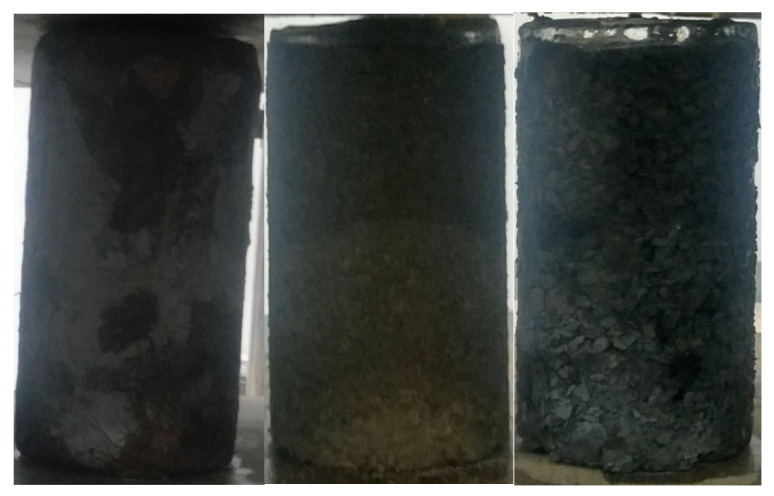
Frozen soil specimens before uniaxial compression tests (clay, sand, and gravel from left to right).

**Figure 3 sensors-22-00885-f003:**
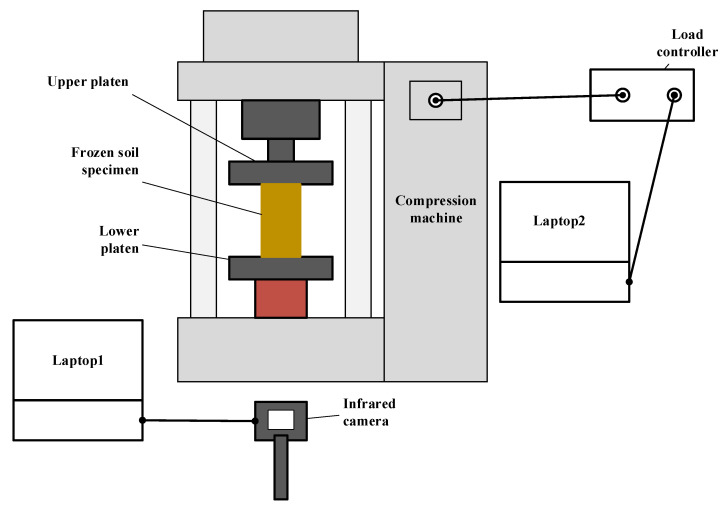
Experimental apparatus setup.

**Figure 4 sensors-22-00885-f004:**
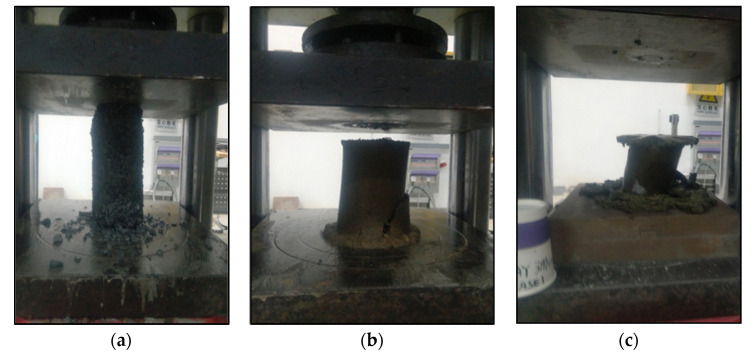
Failure of specimens without cracking behavior: (**a**) gravel; (**b**) sand; and (**c**) clay.

**Figure 5 sensors-22-00885-f005:**
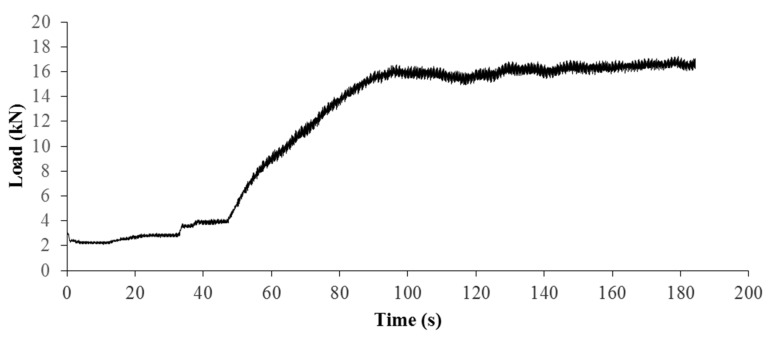
Uniaxial load-time curve of clay specimen.

**Figure 6 sensors-22-00885-f006:**
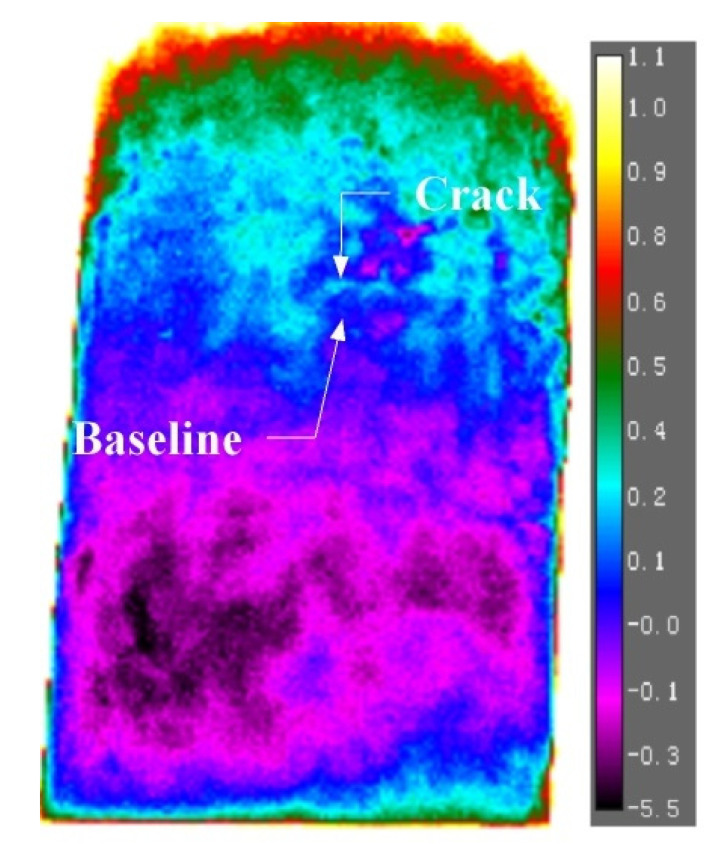
Temperature profile of clay specimen (unit of scale bar is in °C).

**Figure 7 sensors-22-00885-f007:**
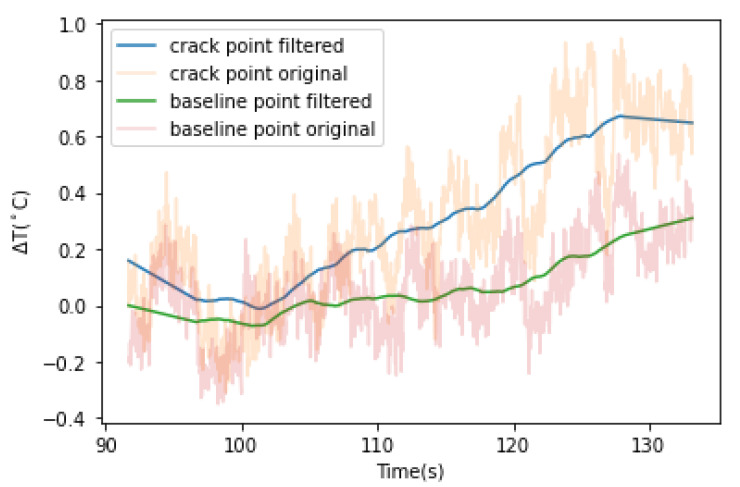
Temperature–time curve of crack point and baseline for clay specimen.

**Figure 8 sensors-22-00885-f008:**
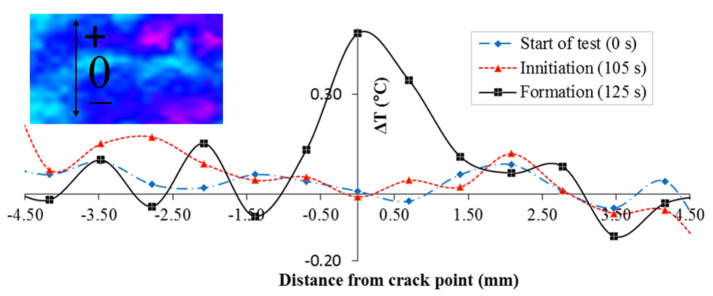
Temperature profile along a measurement line approximately perpendicular to the crack (the position of the measurement line is marked on the temperature profile attached on the top left).

**Figure 9 sensors-22-00885-f009:**
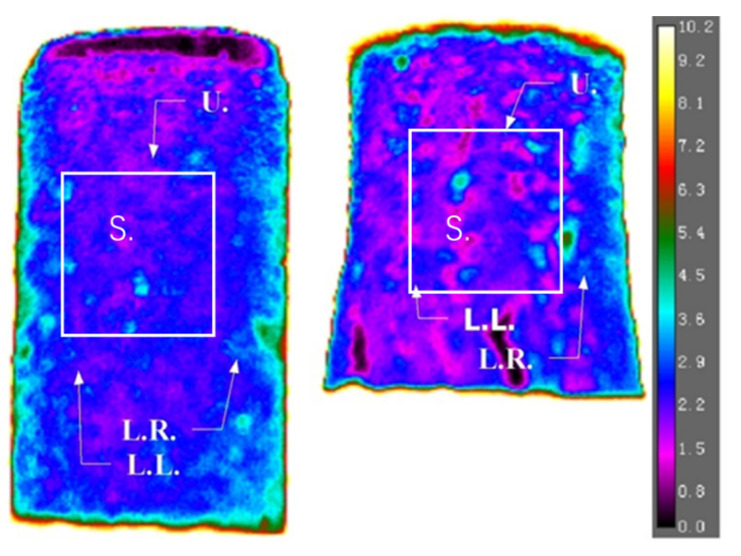
Temperature profile of frozen sand specimen at the start (**right**) and end (**left**) of the compression test (unit of scale bar is in °C).

**Figure 10 sensors-22-00885-f010:**
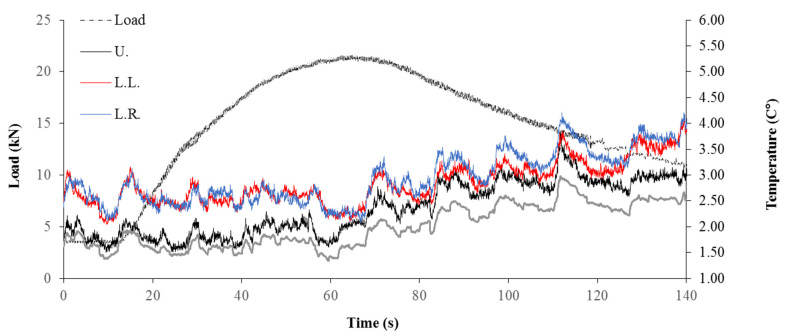
Load curve and temperature–time curve of sand specimen.

**Figure 11 sensors-22-00885-f011:**
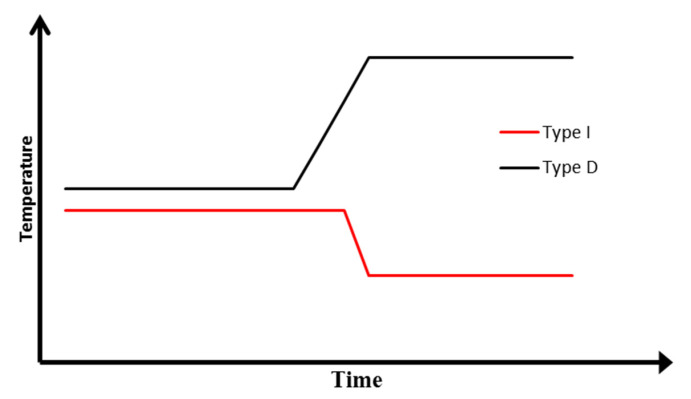
Ideal temperature–time curves of crack type I and type D.

**Figure 12 sensors-22-00885-f012:**
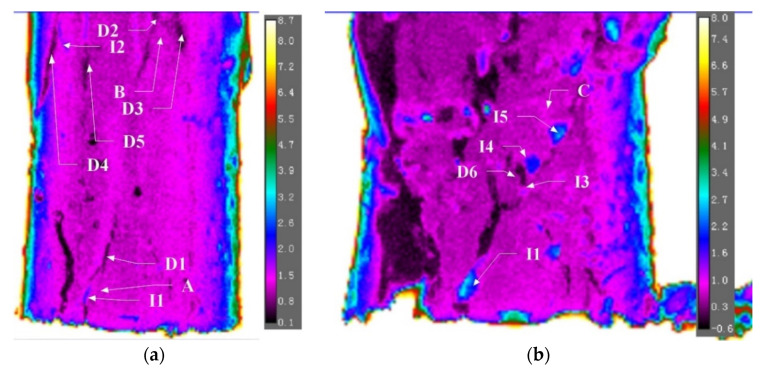
Locations of cracks before plastic deformation (**a**) and after plastic deformation (**b**).

**Figure 13 sensors-22-00885-f013:**
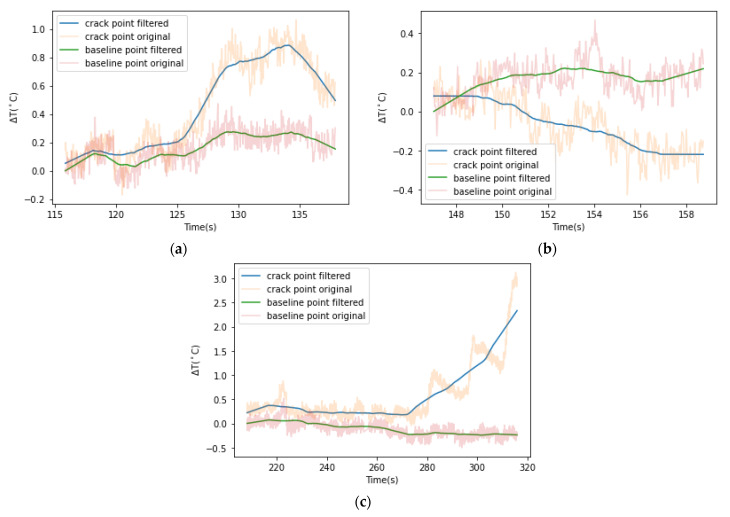
Temporal temperature variation at crack I1 D2 I5 and their immediate baseline A, B, C: (**a**) crack I1; (**b**) crack D2; and (**c**) crack D5.

**Figure 14 sensors-22-00885-f014:**
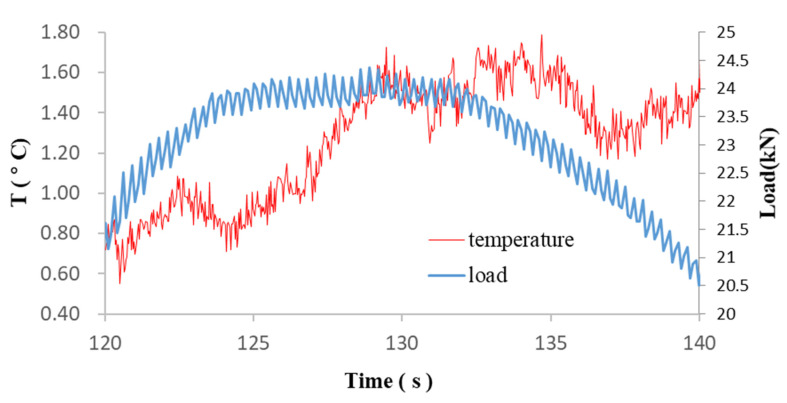
Variation of load and temperature with time at crack point I1.

**Figure 15 sensors-22-00885-f015:**
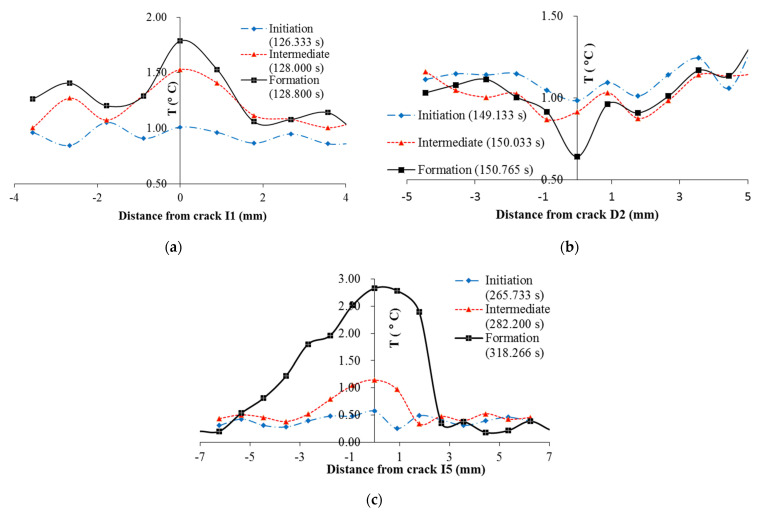
Temperature profile during crack formation: (**a**) crack I1; (**b**) crack D2, and (**c**) crack I5.

**Figure 16 sensors-22-00885-f016:**
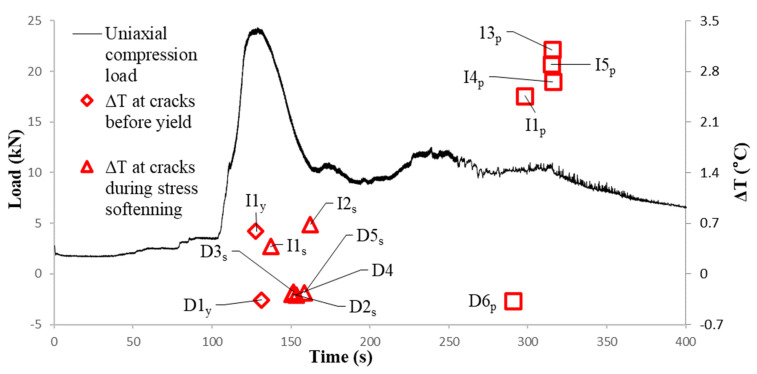
Load–time curve with magnitudes of temperature change at cracks.

**Figure 17 sensors-22-00885-f017:**
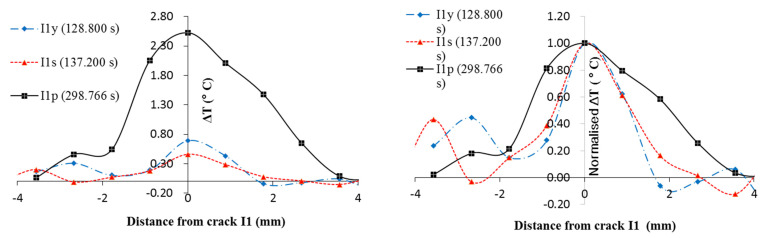
Temperature profile and normalized ΔT at crack I1 at yield, stress softening, and plastic stage.

**Figure 18 sensors-22-00885-f018:**
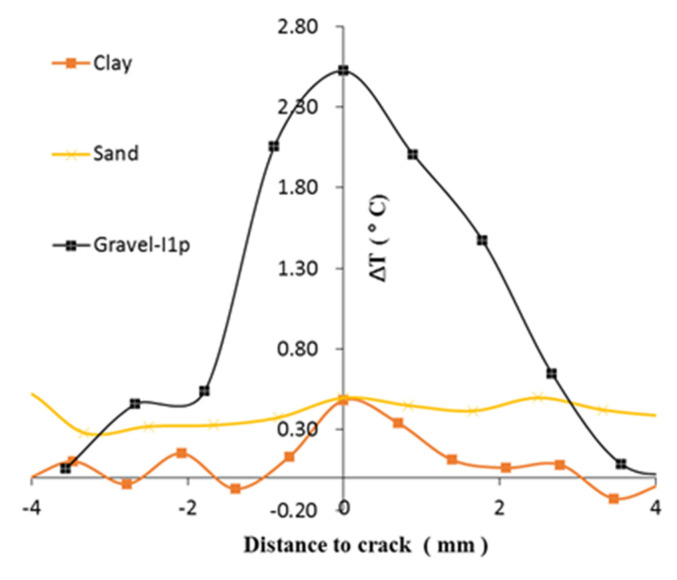
Comparison of temperature profiles across cracks during plastic deformation.

**Figure 19 sensors-22-00885-f019:**
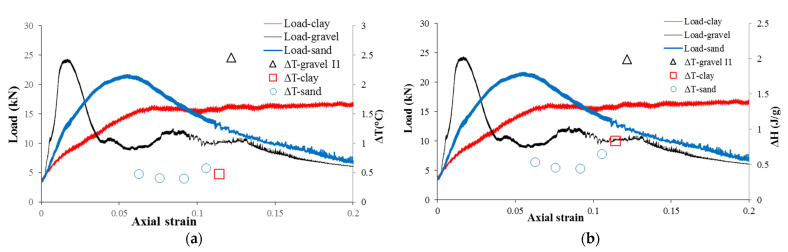
Inter-particle friction in frozen soils: (**a**) comparison of ΔT with stain and load; and (**b**) comparison of ΔH with stain and load.

**Figure 20 sensors-22-00885-f020:**
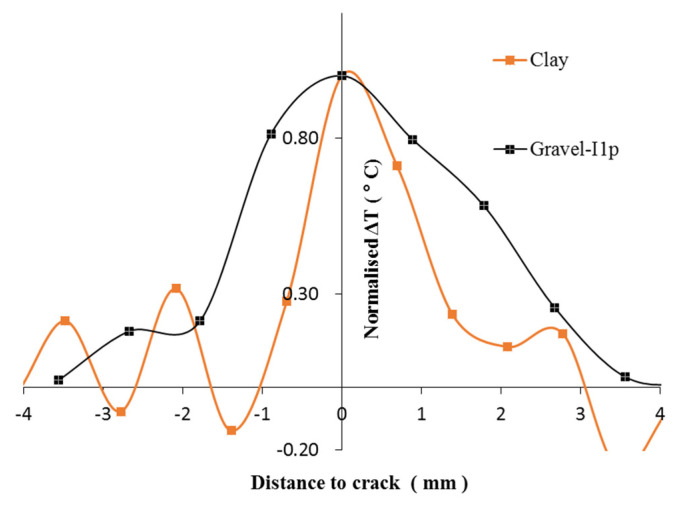
Comparison of ΔT normalised ΔT between clay and gravel cases.

**Table 1 sensors-22-00885-t001:** Properties of specimens (N.M. represents no measurement).

Material	Height (mm)	Diameter (mm)	Weight (kg)	Density (kg/m^3^)	Specific Gravity	Porosity
clay	150	100	N.M.	N.M.	N.M.	N.M.
sand	171	100	2.62	1948.76	2.25	0.24
gravel	179	100	2.75	1952.96	2.51	0.37

**Table 2 sensors-22-00885-t002:** Values of $C_{m}$ from the literature.

Material	$$C_{m}\left(J\cdot {\mathrm{gK}}^{-1}\cdot {{}^{\circ}C}^{-1}\right)$$	Source	Description
Clay	1.750	[22]	Saturated and frozen, measured at 0 °C
Sand	0.770	[23]	Dry, measured at 0 °C
Gravel	0.810		

**Table 3 sensors-22-00885-t003:** Values $C_{m}$ of adopted in this research.

Material	$$C_{m}\;(J/(\mathrm{gK}))$$
Frozen Clay	1.750
Frozen Sand	1.116
Frozen Gravel	0.810

## Data Availability

The data presented in this study are available on request from the corresponding author.

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
