# Peer review of "Crack Detection in Frozen Soils Using Infrared Thermographic Camera"

_sensors, 2022, doi:10.3390/s22030885_

Round 1

Reviewer 1 Report

Dear Authors,

The manuscript presents an experimental work to determine the potential cracks in frozen soils using infrared thermographic camera. It is always very appreciated to apply model techniques for monitoring the potentially induced impacts in subsurface because of the difficulties in direct monitoring methods. However, the manuscript does not clearly state the significance of their work in terms of its field applications and the transformability of the used method as well. Therefore the manuscript is considered as a major revision and the following comments must be addressed before it can be accepted for publication:

  1. It is clearly written that the infrared thermographic camera has been applied in the other fields of detecting failures in structure not the frozen soils. However, how about the fields of the structures above ground, which may have problems when it is frozen? e.g. wind turbines, air crafts? The novelty of the manuscript should be addressed in the way of detecting such a behavior for the soils in the underground not just for the frozen part. Please add literatures for that.
  2. At the end of the manuscript, there is only one sentence to state the difficulties of applying this method into the field. But this is not enough for a scientific publication. Although the transformability is not the topic of the manuscript, it should be discussed clearly in the manuscript to allow the readers to understand the limitations of the applied method, and the uncertainties. Please add one discussion chapter for that.
  3. For a manuscript of experimenting different types of soils, please clearly state all the properties of the used soils in the experimental setup, such as weight, density, porosity, specific gravity and grain size distribution etc., as well as the thermal properties because these all belong to the characteristics of the soils, which are very crucial to understand the experiment.
  4. In Page 4, line 122, if the accuracy is +/- 2°C, plotting temperature profiles with absolute temperatures do not make any sense if the relative change is only about 0.5°C e.g. in figure 7. The thermal sensitivity is much higher at 0.05°C, therefore, all the temperature profiles should be normalized to show the relative change only. Since there is always a fluctuation in your temperature profile, the profile should be thus filtered to reduce the background noises.
  5. Figure 6, 9 and 12, there is no unit in the legend, therefore, the figures are useless to understand the temperature distribution in the sample. If this is simply in °C, please explain why the temperatures in Figure 9 are basically all above 0°C although it is completely frozen.
  6. The manuscript only presents three samples to demonstrate the applicability of the methods. It is not enough to prove the reproducibility. If there were other experiments, please include them into the manuscript and analyze them. If there were none, please repeat your experiments, at least three times for each soil sample. The data should be submitted together with the manuscript as supplementary materials.

Author Response

Thanks for the detailed comments and please find attached the file for the response.

Reviewer 2 Report

In this work, Infrared Thermographic was used to detect the crack in frozen soils. This tool can detect changes in temperature due to inter-particle friction. Phenomena that occur during uniaxial compression tests have been described in detail based on the surface temperature profile of the sample recorded by an infrared video camera. However, some notes need to be added so that the discussion of the article is more comprehensive.  Generally, this manuscript requires minor revisions before being considered for publication in Sensors or any journal in this field.

  1. Caption figure 4 is not clear. Which soil category is shown? Clay, sand, or gravel?
  2. Figure 15 is too crowded. it's preferred to rearrange especially the legend
  3. Statement in line 229 needs the additional data about the load vs temperature for gravel specimens. It stated that “……….. but subsequently decreases due to the decrease of load”.
  4. Graph 17 shows the comparison of temperature profiles across cracks during plastic deformation, it can be highlighted that the temperature change of gravel is higher than that of sand and clay. The discussion will be more comprehensive if it is related to the chemical element content and particle shape in the three specimens

Author Response

(The authors gave the same response as above.)

Reviewer 3 Report

1. There is NO section of method.  Please note this is a serious problem.  Please add it.

2. Please define each term or variable which it appears.

3.  Figure 1 is too simple.  Please note this is the only figure in your manuscript to address the crack sensoring process.

4. Three-dimensional figures may be better than 2-d figures in the manuscript to address the processes of sensoring.

5. In Figure 3, the left has the image of a person.  Please remove the man's imagine.  You may add some text in the captain of the figure, such as "Water was added from the top of the column".

Author Response

(The authors gave the same response as above.)

Round 2

Reviewer 1 Report

Dear Authors,

Thank you a lot for the efforts to improve the manuscript as suggested. On my opinion, it is a very interesting manuscript regarding the field of application. Unfortunately, it still cannot be accepted due to the following reason mainly regarding the experimental design:

In the line 130 of page 4 “Tests were conducted on 6 samples for each soil type as multiple samples for each…” This is really astonishing that only one sample out of 6 samples can be used due to all different kinds of failure. On my opinion, this is a serious flaw in your experimental design. Please clearly discuss the reasons for all these failures happened, suggest possible ways to avoid that and discuss the possibility to improve your experiment. Besides, regarding the experiment with frozen sand, please explain how it can be guaranteed that the sample is still completely frozen although the temperature is all above 0.0°C. The sample was actually just moved out of the freezer and performed a load test with 140 s. It is astonishing that the sample has melted so fast. If the room temperature was the reason, the answer is easy, please find a lab which the room temperature can be maintained at least lower than 30 °C and repeat all the experiments again.

Author Response

The following lines is added in response to the comments. From Line 134 to Line 146:

‘Cracking occurred on either or both sides of specimen by chance, which cannot be practically controlled. Yielding and disintegrating of specimens occurred because the specimens were partially melted. The authors actually attempted to repeat the tests under a lower room temperature (around 15 °C) in the winter, However, satisfactory improvement on avoiding the melting of specimens were not achieved. To avoid the melting of specimens, the compression tests on specimens should be conducted under temperature below freezing point of water following [1-3]. In the previous researches [1-3], the low temperature of specimens during test were maintained by immersing the specimens in liquid cool materials. Unfortunately, such cooling system was not available to the authors and forbids the direct observation of the infrared thermographic camera on the specimens during compression tests. Therefore, all specimens were not guaranteed completely frozen during our tests.’

  1. Bragg, R.A.; Andersland, O.B. Strain rate, temperature, and sample size effects on compression and tensile properties of frozen sand. Engineering Geology 1981, 18, 35-46, doi:https://doi.org/10.1016/0013-7952(81)90044-2.
  2. Baker, T.; Jones, S.; Parameswaran, V. Confined and unconfined compression tests on frozen sand. 2010.
  3. Liu, X.-z.; Liu, P. Experimental research on the compressive fracture toughness of wing fracture of frozen soil. Cold Regions Science and Technology 2011, 65, 421-428, doi:https://doi.org/10.1016/j.coldregions.2010.11.006.

Reviewer 3 Report

good job.

Author Response

Many thanks for your review.